# Anti-Inflammatory and Pro-Differentiating Properties of the Aryl Hydrocarbon Receptor Ligands NPD-0614-13 and NPD-0614-24: Potential Therapeutic Benefits in Psoriasis

**DOI:** 10.3390/ijms22147501

**Published:** 2021-07-13

**Authors:** Giorgia Cardinali, Enrica Flori, Arianna Mastrofrancesco, Sarah Mosca, Monica Ottaviani, Maria Lucia Dell’Anna, Mauro Truglio, Antonella Vento, Marco Zaccarini, Christos C. Zouboulis, Mauro Picardo

**Affiliations:** 1Laboratory of Cutaneous Physiopathology and Integrated Center of Metabolomics Research, San Gallicano Dermatological Institute, IRCCS, 00144 Rome, Italy; giorgia.cardinali@ifo.gov.it (G.C.); enrica.flori@ifo.gov.it (E.F.); arianna.mastrofrancesco@ifo.gov.it (A.M.); sarah.mosca@ifo.gov.it (S.M.); monica.ottaviani@ifo.gov.it (M.O.); marialucia.dellanna@ifo.gov.it (M.L.D.); mauro.truglio@ifo.gov.it (M.T.); antonella.vento@ifo.gov.it (A.V.); 2Genetic Research, Molecular Biology and Dermatopathology Unit, San Gallicano Dermatological Institute, IRCCS, 00144 Rome, Italy; marco.zaccarini@ifo.gov.it; 3Departments of Dermatology, Venereology, Allergology and Immunology, Dessau Medical Center, Brandenburg Medical School Theodore Fontane and Faculty of Health Sciences Brandenburg, 06847 Dessau, Germany; christos.zouboulis@gmx.de

**Keywords:** skin, aryl hydrocarbon receptor, inflammation, keratinocyte differentiation, psoriasis

## Abstract

The aryl hydrocarbon receptor (AhR), a ligand-activated transcription factor expressed in all skin cell types, plays a key role in physiological and pathological processes. Several studies have shown that this receptor is involved in the prevention of inflammatory skin diseases, e.g., psoriasis, atopic dermatitis, representing a potential therapeutic target. We tested the safety profile and the biological activity of NPD-0614-13 and NPD-0614-24, two new synthetic AhR ligands structurally related to the natural agonist FICZ, known to be effective in psoriasis. NPD-0614-13 and NPD-0614-24 did not alter per se the physiological functions of the different skin cell populations involved in the pathogenesis of inflammatory skin diseases. In human primary keratinocytes stimulated with tumor necrosis factor-α or lipopolysaccharide the compounds were able to counteract the altered proliferation and to dampen inflammatory signaling by reducing the activation of p38MAPK, c-Jun, NF-kBp65, and the release of cytokines. Furthermore, the molecules were tested for their beneficial effects in human epidermal and full-thickness reconstituted skin models of psoriasis. NPD-0614-13 and NPD-0614-24 recovered the psoriasis skin phenotype exerting pro-differentiating activity and reducing the expression of pro-inflammatory cytokines and antimicrobial peptides. These data provide a rationale for considering NPD-0614-13 and NPD-0614-24 in the management of psoriasis.

## 1. Introduction

The aryl hydrocarbon receptor (AhR), also called the dioxin receptor, is a cytosolic ligand-activated transcription factor widely expressed in all skin cell types [1]. It exerts a crucial role in skin integrity and immunity by regulating several genes involved in epidermal differentiation and cutaneous homeostasis [2,3,4,5,6,7]. Many studies evaluate how the dysregulation of the AhR signaling pathway can be involved in the pathogenesis of inflammatory skin diseases and tumors and, consequently, these aspects make AhR particularly interesting as a possible therapeutic target [7,8,9,10,11]. The AhR is activated by UVB exposure, different endogenous and exogenous molecules such as environmental toxins, polycyclic aromatic hydrocarbons (PAHs), microbial bioproducts, photo-induced chromophores, and phytochemicals [6,12,13,14,15]. A potent activator of AhR, the 2,3,7,8-tetrachlorodibenzo-p-dioxin (TCDD), develops in humans chloracne, a severe form of acne associated with epidermal hyperproliferation and hyperkeratinization and with up-modulation of genes related to the cornified envelope [16,17,18,19,20,21]. Several studies led to the identification of a canonical and non-canonical AhR signal transduction pathways [10,22,23]. In the canonical one, following the binding with the agonists, AHR translocates into the nucleus inducing the transcription of multiple responsive genes, including the xenobiotic-metabolizing enzymes of the cytochrome P450 family (CYP1A1 and CYP1B) and Blimp1 [24]. Through the non-canonical pathway AhR can interact with other signaling cascades [25,26] related to cell growth/differentiation, inflammatory response, and anti-oxidant defense such as epidermal growth factors receptor (EGFR), down-stream mitogen-activated protein kinases (MAPK) [27,28], NF-kB [7,29] STATs [30,31], and nuclear factor-erythroid 2-related factor 2 (NRF-2) [32,33,34]. Some studies demonstrated that AhR antagonists can prevent skin cancer, whereas the agonists can be useful for the treatment of inflammatory skin diseases [7,8]. The topical treatment of lesional skin with AhR agonists demonstrated beneficial effects in AD and psoriasis [8,35,36,37]. Among the endogenous ligands the 6-formylindolo [3,2-b] carbazole (FICZ), a tryptophan metabolism derivative obtained through UV or visible light exposure [38], effectively resulted in rescuing the impaired keratinocyte terminal differentiation and dampening inflammation in psoriatic skin biopsies and in imiquimod-induced mouse models of psoriasis. In contrast, the use of AhR antagonists increased inflammation and induced the expression of psoriasis-associated genes in non-lesional skin [8,37]. Therefore, the identification of new classes of AhR ligands able to exert beneficial effects on skin diseases without interfering with the expression of genes involved in side effects is an important research goal. Recently, Marafini et al. tested the anti-inflammatory properties of new chemical compounds, structurally related to FICZ, designed as AhR ligands. To discover the new chemical entities (NCEs) they used an in silico scaffold hopping approach starting from the lowest energy conformation of b-carboline derivatives (shown in vitro to increase interleukin 22 (IL-22) levels and decrease interferon γ (IFN-γ) level in peripheral blood mononuclear cells stimulated with a aCD3/aCD28 antibody [39], and replacing the central b-carboline core. Upon in silico ADMET screening, NCEs were synthetized for in vitro screening. In a model of inflammatory bowel diseases (IBDs), these AhR ligands were able to down-regulate the inflammatory signals and improve TNBS-induced colitis without signs of toxicity and adverse side effects [40]. Since psoriasis and IBDs are both chronic diseases based on a persistent inflammatory state that can be counteracted using similar biological drugs, we have extended the previous data from Marafini et al. testing these AhR ligands on two different three-dimensional (3D) models of psoriasis: a reconstructed human epidermal equivalent (HEE) and a full-thickness reconstructed skin which represents a more complex system due to the presence of psoriatic fibroblasts. We compared their effects with those induced by the natural AhR ligand FICZ. Moreover, to take into account the complex cutaneous paracrine network that controls skin homeostasis and inflammatory process, we tested the molecules on different skin cell populations such as normal human keratinocytes, melanocytes, and sebocytes. Taken together, our results provide a rationale for further preclinical/clinical studies to evaluate the possible use of the aforementioned molecules for the treatment of skin diseases characterized by an inflammatory state associated with an impaired keratinocyte terminal differentiation such as psoriasis.

## 2. Results

### 2.1. NPD-0614-13 and NPD-0614-24 Activated AhR Signaling in NHKs

We investigated the ability of the compounds to activate the AhR signaling pathway examining the mRNA expression of AhR target genes, such as NRF-2, CYP1A1, CYP1B and NAD(P)H Quinone dehydrogenase 1 (NOQ1) after treatment with NPD-0614-13 and NPD-0614-24 at different doses (6–12, 5–25 and 50 μM) and time points (3–6–24 h). The two molecules, starting from the dose of 12.5 μM, determined an early up-regulation of NRF-2 mRNA level (3–6 h), a gradual induction of CYP1A1 and CYP1B mRNA, more significant at the higher doses, and a late up-regulation of NQO1 mRNA expression (24 h) (Figure 1b). For both compounds, the 25 μM dose was the most effective, and hence was selected for the subsequent experiments. As expected, the stimulation with FICZ induced a rapid induction of CYP1A1 and CYP1B mRNA (Appendix A). To confirm the AhR activation by the new molecules we evaluated the nuclear translocation of the AhR by immunofluorescence analysis after treatment with NPD-0614-13 and NPD-0614-24. The results revealed that, in untreated cells, AhR is mainly localized in the cytoplasm. The treatment with both compounds caused an increase in nuclear fluorescence signals (arrows), indicating the nuclear translocation of the activated receptor (Figure 1c). To confirm that the compounds interact with AhR, we investigated the mRNA expression of AhR gene targets (CYP1A1, CYP1B1) in NHKs transiently transfected with siRNA for AhR (siAhR) or control (siCtr). Both molecules significantly increased CYP1A1 and CYP1B1 gene expression in siCtr cells, but failed to induce them in AhR-deficient NHKs (Appendix A).

### 2.2. NPD-0614-13 and NPD-0614-24 Safety Profile in Different Skin Cell Populations

In addition to keratinocytes, the AhR system is expressed in several skin cells such as fibroblasts, melanocytes, mast cells and sebocytes [4]. We analyzed the biological effects of the molecules in the different skin cell populations evaluating their capacity to act by physiological mechanisms in absence of toxic adverse events. In SZ95 sebocytes we looked for the appearance of typical biomarkers induced by 2,3,7,8-tetrachlorodibenzo-p-dioxin (TCDD), the chloracnegenic AhR ligand. TCDD reduces the expression of K7 (a marker of sebocytes differentiation) and increases the expression of K10 (a marker of keratinocyte differentiation), resulting in a switch towards keratinocyte-like differentiation [11,41]. The exposure of SZ95 sebocytes to NPD-0614-13, NPD-0614-24 did not induce any modification of the expression of K7, nor an increase in the expression of K10 (Figure 2a). Furthermore, no changes in lipogenesis (induction or inhibition) or variations in the percentage composition, in terms of saturated, mono- and polyunsaturated fatty acids were detected (Figure 2b). Chloracne is also characterized by an alteration of the proliferative and differentiation processes of keratinocytes, determining the thickening of the squamous epithelium and the metaplasia and hyperkeratinization of the ducts of the sebaceous gland. The production of reactive oxygen species (ROS) is a critical step in these events [42,43,44,45]. The treatment of NHKs with NPD-0614-13, NPD-0614-24 neither affected cellular ROS levels (0.91 ± 0.01, 0.96 ± 0.09 and 1.02 ± 0.05 vs. control, respectively), nor altered the expression of proteins involved in cell proliferation or differentiation, such as p53 and phospho-EGFR (Figure 2c), filaggrin and involucrin (Figure 2d), respectively.

In some cases, AhR activation can be associated with skin xerosis and hyper-pigmentation [46,47]. On primary melanocytes (NHMs), the treatment with both molecules neither induced changes in cell dendriticity, a clear sign of melanocyte differentiation, nor in the expression of tyrosinase, a key protein in melanogenesis. Accordingly, the intracellular melanin content was not modified (Figure 2e), indicating that the compounds, at the concentration used, did not affect the pigmentation process.

### 2.3. NPD-0614-13 and NPD-0614-24 Recovered the Alteration of Cell Proliferation Induced by LPS or TNF-α

The activation of MAPKs, such as phosphoERK1/2, as well as the degradation of the cell cycle protein p21WAF1/CIP1 (p21), are associated with an altered proliferation rate in keratinocytes activated by pro-inflammatory stimuli [48]. In order to verify whether NPD-0614-13 and NPD-0614-24 were able to interfere with these intracellular pathways, NHKs were stimulated with TNF-**α** (20 ng/mL) or LPS (10 μg/mL). As expected [48], an increased phosphorylation of ERK1/2 and a degradation of p21 was observed. NPD-0614-13 and NPD-0614-24 resulted effective in reducing the levels of phospho-ERK1/2 and inhibiting the degradation of p21 (Figure 3a,b), thus recovering the altered proliferation. These results were confirmed by the reduction of Ki67 positive cells observed by immunofluorescence analysis (Figure 3c).

### 2.4. NPD-0614-13 and NPD-0614-24 Counteracted the Pro-Inflammatory Effects Induced by LPS or TNF-α

An increased amount of intracellular ROS and the activation of p38 MAPK and AP-1/c-Jun are known to play an important role in the production of pro-inflammatory mediators in TNF-α or LPS-activated NHKs [49,50]. To evaluate the anti-inflammatory properties of the molecules, NHKs were stimulated with TNF-α (20 ng/mL) or LPS (10 μg/mL) for 1 h. NPD-0614-13 and NPD-0614-24 suppressed the increase of ROS levels (Figure 4a) and significantly reduced phospho-p38 and AP-1/c-Jun activation (Figure 4b). Then, we analyzed the phosphorylation and nuclear translocation of NF-kB, another master regulator of inflammatory response, after treatment with LPS or TNF-α, in the presence or absence of the compounds. Both molecules reduced the NF-kB p65 phosphorylation in cytoplasmic and nuclear fractions (Figure 4c). The immunofluorescence analysis showed a lower number of cells with nuclear phospho-NF-kBp65 (arrows), in the presence of NPD-0614-13, NPD-0614-24 (Figure 4c). Furthermore, the treatment with NPD-0614-13, NPD-0614-24 significantly decreased IL-6 and IL-8 mRNA and protein levels comparably or even more efficiently than FICZ (Figure 5a,b). To confirm that the anti-inflammatory activity of the compounds was related to the interaction with AhR, we evaluated IL-6 and IL-8 release after TNF-α or LPS treatment in NHKs transiently transfected with siRNA for AhR (siAhR) or control (siCtr). NPD-0614-13 and NPD-0614-24 significantly decreased the amount of cytokines released in siCtr cells, but almost completely failed to modify cytokine expression in siAhR cells (Appendix A).

### 2.5. NPD-0614-13 and NPD-0614-24 Showed Pro-Differentiative and Anti-Inflammatory Effects in Psoriasis-Like Human Epidermal Equivalents (PS-HEE) and Reconstructed Human Psoriatic Skin Equivalents (PSE)

Based on the encouraging results related to the anti-inflammatory action induced in NHKs by the two molecules, we investigated their possible effects in a psoriatic human epidermal equivalent model (PS-HEE) using a human keratinocyte immortalized cell line (Ker-CT). We tested both molecules on HEEs maintained in basal culture conditions to detect any possible side effects on the morphology of the skin and the expression of differentiation and inflammation markers. In specimens stained with hematoxylin–eosin (H&E) no modification of the epidermal architectural organization of the keratinocyte layers was observed under treatment with the two compounds (Figure 6a). NPD-0641-13 and NPD-0614-24 did not modify two parameters altered in chloracne, i.e., the thickness of both stratum corneum (SC) and viable cell layers of the epidermis (VE), compared to control HEEs (Figure 6b). By contrast, FICZ induced mild signs of stratum corneum thickening and overall epidermal thinning (Figure 6b). Qualitative and quantitative analyses of the immunofluorescence staining demonstrated that NPD-0641-13 and NPD-0614-24 did not affect filaggrin expression (Figure 6c,d). The analysis of mRNA expression showed no significant changes in the level of filaggrin and caspase 14, a crucial enzyme involved in filaggrin maturation (Figure 6e). Moreover, the substances were not able to induce the mRNA expression of psoriasin (S100A7) and beta-defensin 2, two well-known anti-microbial peptides up-modulated in psoriatic skin [51,52,53] (Figure 6e). To induce a psoriatic phenotype, we stimulated HEEs with a combination of Th1/Th17 cytokines including TNF-α, IL-1α, and IL-17A. H&E staining on PS-HEE sections demonstrated some typical psoriatic skin abnormalities, such as thickening of the stratum corneum and the loss of the granular layer (Figure 7a). NPD-0614-13 and NPD-0614-24 partially recovered the alterations induced by cytokines, improving epidermal morphology with a reduction of the stratum corneum thickness (Figure 7b). However, either FICZ or derived molecules did not rescue the epidermal thinning. Qualitative and quantitative analyses of the immunofluorescence staining performed on serial sections of HEEs and PS-HEEs revealed that the stimulation with pro-inflammatory cytokines reduced the expression of the late differentiation markers involucrin and filaggrin (Figure 7c,d). NPD-0614-13 and NPD-0614-24 were effective in counteracting these effects. We observed an increased expression of involucrin on the suprabasal layers and of filaggrin on the stratum spinous and the stratum corneum compared to the untreated PSO-HEEs (Figure 7c,d). FICZ treatment, compared with derived molecules, showed a similar effect on recovering filaggrin expression, but did not rescue the reduced expression of involucrin (Figure 7c,d). Furthermore, we evaluated the modulation induced by NPD-0614-13 and NPD-0614-24 on the expression of S100A7 and beta-defensin 2. Immunofluorescence analysis revealed a significant increase in the expression of both peptides after stimulation with Th1/Th17 cytokines. The treatments with both compounds resulted in a reduction of psoriasin and, to a lesser extent, beta-defensin 2 expression (Figure 7c,d). Parallel analysis of the mRNA expression levels confirmed the immunofluorescence staining evidencing a more pronounced activity of FICZ on reducing the antimicrobials peptides transcripts (Figure 7e). Moreover, we analyzed the anti-inflammatory role of NPD-0614-13 and NPD-0614-24 by evaluating the RNA transcripts for IL-1α, IL-1β, IL-6, and IL-8. The molecules significantly reduced the mRNA expression of all pro-inflammatory cytokines (Figure 7f). In addition, we proceeded to explore the beneficial effects in a 3D full-thickness human skin model that better mimics psoriatic skin for the presence of fibroblasts harvested from lesions, which have been previously shown to exert a relevant role in pathogenesis of psoriasis [54,55,56,57,58,59]. The treatment with both compounds enhanced the differentiation process as assessed by the epidermal thinning associated with stratum corneum thickening and by the increased expression of filaggrin. Both molecules were also effective in reducing the expression of beta-defensin-2 and the release of pro-inflammatory cytokines IL-6 and IL-8 (Figure 8a–c). The results observed with FICZ treatment demonstrated that the natural ligand, although exerting similar anti-inflammatory effects, was less effective than the derivatives in recovering the psoriatic skin phenotype at both epidermal and stratum corneum level (Figure 8a–c).

## 3. Discussion

AhR was initially considered to be mainly involved in the metabolism of environmental chemicals. However, recent research investigates in the role of AhR in immune functions, particularly in the realm of inflammation [29]. The type of AhR ligand and the cell context are important factors that can differently modulate the AhR-driven signals acting as initiators or attenuators of inflammatory processes, hence explaining the contradictory literature data [27,60,61,62,63]. NPD-0614-13 and NPD-0614-24, two structurally related FICZ molecules, were tested by Marafini et al. [40] NPD-0614-13 and NPD-0614-24 in an experimental mouse model of TNBS-induced colitis. The authors demonstrated their ability to ameliorate the ongoing colitis through the induction of IL-22 and the down-regulation of inflammatory cytokines by targeting immune cells in absence of adverse events. Based on these results on immune cells, we focused our study on skin cell populations, taking into account the crucial role of the complex paracrine network among the different cutaneous cell types. We aimed to investigate the potential benefits of using these compounds in inflammatory skin diseases, such as psoriasis, by the regulation of epidermal differentiation and inflammation, without interfering with physiological skin functions. NPD-0614-13 and NPD-0614-24, at the concentrations used, did not directly modify the physiological functions of sebocytes, melanocytes, and keratinocytes, showing a similar biological activity with the endogenous AhR ligand FICZ. FICZ is a substrate for the metabolizing activity of CYP1A1 enzyme [9]. An important aspect is that both compounds induced CYP1A1 and CYP1B1 to a lesser extent with respect to FICZ, suggesting the induction of a milder detoxifying response. This event could be an index that both molecules, although maintaining an anti-inflammatory capacity similar to FICZ, showed a higher safety profile. This aspect associated with the absence of the adverse event pattern typical of several environmental pollutants, can be clinically relevant.

In response to several pro-inflammatory stimuli, human keratinocytes activate intracellular signaling mediators, such as p38 MAPK, involved in the activation of NF-kB and c-Jun/AP-1, key regulators of genes for inflammatory cytokines [49,50,64]. NPD-0614-13 and NPD-0614-24 resulted effectively in reducing the phosphorylation levels of p38 and NF-kBp65 and thus the release of pro-oxidative cytokines. Several studies demonstrated that AhR agonists can reduce the inflammatory response in keratinocytes and a mouse model of skin inflammation controlling transcription factors of the AP-1 family (i.e., c-Jun and JunB) [8,65]. AP-1 family represents a dynamic network of transcription factors whose regulation affects epidermal homeostasis and the skin inflammatory process and are dysregulated in psoriasis [8,66,67]. The efficacy of NPD-0614-13 and NPD-0614-24 in reducing the phosphorylation of c-Jun, a member of AP-1 family induced by pro-inflammatory stimuli, confirmed their ability to counteract the inflammation via a physiological mechanism of AhR activation. Overall, these results suggest that both compounds are interesting from a clinical point-of-view due to their regulatory activity on upstream and multipotent signal mediators. Thus, they can be considered to counteract inflammatory conditions induced by different stimuli such as TNF-α and LPS that trigger multiple signaling pathways by interacting with the specific membrane receptors TNFR or Toll-like receptor 4, respectively [48]. In addition, NPD-0614-13 and NPD-0614-24 reduced the amount of intracellular ROS through the activation of NRF-2, which is a master switch of the antioxidant response by upregulating the expression of antioxidative enzymes (i.e., HMOX1, NQO1, and Glutathione S-transferase). These results match with literature data showing a cross-talk between AhR signaling and NRF-2 [68,69].

Complex interactions between dermal/epidermal cells and the inflammatory infiltrate are involved in the pathogenesis of psoriasis. In psoriatic skin, there is a dysregulation of genes implicated in the catabolism of tryptophan and consequently of naturally derived products, such as FICZ [70,71] leading to a decreased activation of AhR and an increased expression of inflammatory mediators. Hence, it is important to study new AhR activators able to mimic the beneficial anti-inflammatory and pro-differentiative effects, which are exerted physiologically by natural ligands, considering AhR as a valid target for therapeutic approaches in inflammatory skin diseases. Three-dimensional skin models may provide better information on skin disease phenotypes and let the effects on keratinocyte differentiation across epidermal layers been studied in a more relevant manner. Indeed, recent advances and optimization of three-dimensional skin models provide an attractive and promising alternative for pre-clinical research [72,73,74,75,76]. To evaluate the possible benefits of NPD-0614-13 and NPD-0614-24 in the management of psoriasis, we engaged two different three-dimensional models characterized by different levels of complexity. The first one takes into account the pathogenic role of pro-inflammatory cytokines that are able to induce transcriptome alterations similar to those observed in psoriatic skin [77]. The second system represents an optimized model of psoriasis that better mimics the in vivo psoriatic skin phenotype due to the presence of fibroblasts isolated from psoriatic skin lesions. Some studies demonstrated significant changes in the proteomic profile of psoriatic fibroblasts responsible for the upregulation of growth factors, bioactive mediators, and pro-inflammatory cytokines which promote and sustain the hyperproliferation of keratinocytes and the inflammatory status [55,57,59,73]. In 3D systems maintained in basal conditions, FICZ, differently from our molecules, induced mild signs of chloracne phenotype such as the thickening of the stratum corneum and the epidermal thinning. Moreover, NPD-0614-13 and NPD-0614-24 improved epidermal morphology through the recovery of the psoriatic skin phenotype and stimulated effectively the epidermal differentiation and the anti-inflammatory response. Interestingly, the comparison of the results observed in the two psoriasis models suggests that the molecules can activate beneficial effects against an inflammatory state induced either by direct cytokine stimulation or by a more complex paracrine network regulated by the dermal compartment. The comparison with the natural ligand FICZ revealed that NPD-0614-13 and NPD-0614-24 have similar properties in activating the anti-inflammatory response; however, the new AhR ligands better recovered psoriasis skin phenotype by reducing more effectively keratinocyte hyperproliferation and by enhancing the cornification process. No model exists that can completely match psoriatic skin and resume disease pathogenesis and a common limit of some of them is the lack of the contribution of the immune cells. Further studies using in vivo models that take into account the immune axis will allow to deepen the biological mechanisms involved in AhR signaling and specially to validate the efficacy of new molecules for the treatment of psoriasis.

Overall, our data showed that the two molecules exerted similar effects demonstrating that their different structure does not modify their biological activity. Hence, future in vivo pharmacokinetic studies will allow to highlight possible differences between the two compounds, i.e., a greater tissue permanence capacity without signs of toxicity or alteration of skin physiology. In conclusion, this current work identified NPD-0614-13 and NPD-0614-24 as two apparently safe AhR ligands with beneficial effectiveness on targeting crucial biological endpoints in psoriatic skin. The identification of new molecules that exploit physiological mechanisms to activate the AhR pathway may open new therapeutic perspectives to dampen the severity of chronic inflammatory skin diseases.

## 4. Materials and Methods

### 4.1. Materials

M154 defined medium, human keratinocyte growth supplements (HKGS), M254 defined medium, human melanocyte growth supplements (HMGS), fetal bovine serum (FBS), L-glutamine, penicillin/streptomycin, trypsin/EDTA, and D-PBS were purchased from Invitrogen Technologies (Monza, Italy). Sebomed^®^ basal medium was purchased from Merck-Biochrom (Berlin, Germany). Recombinant human epidermal growth factor was purchased from Invitrogen (Milan, Italy). β-tubulin antibody (T7816), GAPDH antibody (G9545), 6-formylindolo (3,2-b) carbazole (FICZ) and lipopolysaccharide (LPS) were from Sigma-Aldrich (Milan, Italy). Aurum^TM^ Total RNA Mini kit, SYBR Green PCR Master Mix, Bradford reagent were from Bio-Rad (Milan, Italy). RevertAid^TM^ First Strand cDNA synthesis kit, si-AhR, (si-AhR, s1200), IL-6 and IL-8 ELISA kit, NE-PER nuclear and cytoplasmic extraction reagents were from Thermo Fisher Scientific (Monza, Italy). Non-specific siRNA (sc-44234) and tyrosinase antibody (sc-7833) were from Santa Cruz Biotechnology (Santa Cruz, CA, USA). The Amaxa^®^ human keratinocyte Nucleofector kit was from Lonza (Basel, Switzerland). The antibodies for p21 Waf1/Cip1 (#2947), phospho-p44/42 MAPK (ERK1/2) (Thr202/Tyr204) (#4370), phospho-p38 MAP kinase (Thr180/Tyr182) (#4511), phospho-EGFR (Tyr1068) (#3777), phosphor-c-JUN (Ser73) (#3270), Keratin 7 (#4465), secondary anti-mouse IgG HRP-conjugated antibody and anti-rabbit IgG HRP-conjugated antibody were purchased from Cell Signaling (Danvers, MA, USA). The anti-NF-κB p65 (phosphoS536) antibody (ab86299), anti-NF-κB p65 antibody (ab32536), anti-involucrin antibody (ab53112), anti-Filaggrin antibody (ab24584), anti-cytokeratin 10 antibody (ab76318), anti-aryl hydrocarbon receptor antibody (ab2769) and rabbit polyclonal to Ki67 proliferation marker were purchased from Abcam (Cambridge, UK). Amersharm ECL Western Blotting Detection Reagent was from GE Healthcare (Buckinghamshine, UK). Protease inhibitor cocktail was from Roche (Mannheim, Germany). Recombinant Human Tumor necrosis factor-α (TNF-α) was from R&D System (Minneapolis, MN, Canada). The antibody anti-p53 was from DakoCytomation (DO-7, Glostrup, Denmark).

### 4.2. New Chemical Ligands

Among the different FICZ derivatives kindly supplied by PPM Services SA-A Nogra Group Company, we selected NPD-0614-13 and NPD-0614-24 (Figure 1a) for a more pronounced anti-inflammatory activity. Molecular docking was performed using SwissDock, a protein-small molecule docking web service based on EADock DSS by the Molecular Modeling group of the Swiss Institute of Bioinformatics. The binding affinity as estimated by the binding free energy between the different ligands and the receptor, demonstrated that NPD-0614-13 and NPD-0614-24 are comparable to the high-affinity natural AhR ligand FICZ. The ligand–protein binding energies for NPD-0614-13, NPD-0614-24, and FICZ were −7.520 kcal/mol, −7.510 kcal/mol, and −7.701 kcal/mol, respectively.

### 4.3. Cell Culture

Normal human keratinocytes (NHKs) and normal human melanocytes (NHMs) were isolated from neonatal foreskin in accordance with a previously described procedure [78]. NHKs were maintained at 37 °C under 5% CO_2_ in the defined medium M154 with HKGS, L-glutamine (2 mM), penicillin/streptomycin (100 µg/mL). NHKs were subcultivated once a week and experiments carried out in cells between passages 2 to 4. NHMs were grown at 37 °C under 5% CO_2_ in M254 medium with HMGS, L-glutamine (2mM), penicillin/streptomycin (100 µg/mL) and used between passage 2 and 10. Immortalized human SZ95 sebocytes, showing morphologic, phenotypic and functional characteristics of normal human sebocytes [79], were maintained in Sebomed^®^ basal medium, supplemented with 10% FBS, L-glutamine (2mM), penicillin/streptomycin (100 µg/mL), recombinant human epidermal growth factor (5 ng/mL) and CaCl_2_ (1 mM) in a humidified atmosphere containing 5% CO_2_ at 37 °C. Cell cultures were routinely tested for Mycoplasma infection. The Institute’s Research Ethics Committee (IFO) approval was obtained to collect samples of human material for research. The study was conducted according to the Declaration of Helsinki Principles. Patients gave written informed consent. For each experiment at least three different donors were used. Cells were plated and 24 h later were stimulated with chemicals in fresh medium, in accordance to the experimental design.

### 4.4. Morphologic Analysis by Inverted Phase Contrast Microscope

Living cell cultures were observed daily and images were captured by an inverted phase microscope (Axiovert 40C, Zeiss, Oberkochen, Germany) equipped with a digital camera.

### 4.5. Human Epidermal Equivalents (HEEs) and Psoriasis-Like HEE (PS-HEEs) Preparation

The immortalized human keratinocyte cell line Ker-CT (ATCC^®^ CRL-4048^TM^) was used to generate 3D human epidermal equivalents (HEE) and 3D psoriasis-like human epidermal equivalents (PS-HEE) [80]. Ker-CT were seeded on 48 cell culture inserts (Thermo Scientific, Roskilde, Denmark; 0.4 μm pore size; 2 × 10^5^ cells per insert) and maintained for seven days in a submerged condition in CnT-Prime Epithelial Culture Medium (CnT-PR) (CellnTEC, Bern, Switzerland) and switched in CnT-Prime 3D Barrier Medium (CnT-PR-3D) in an air–liquid condition for 10 days. The medium was replaced every alternate day. For the psoriasis models, Th1/Th17 cytokines, TNF-α (5 ng/mL), IL-6 (5 ng/mL), IL-1α (10 ng/mL), and Th17A (10 ng/mL) (Peprotech, Cranbury, NJ, USA) were added alone for the first 24 h and then, for the last 3 days of air–liquid interface culture condition, in the absence or presence of the different molecules. At day 10, HEEs and PS-HEEs were processed for gene expression analysis or formalin-fixed and paraffin-embedded for hematoxylin and eosin (H&E) staining, morphometry, and immunofluorescence analysis. Supernatants were collected for cytokine evaluation by ELISA assay.

### 4.6. D Full-Thickness Human Psoriasis Skin Equivalent (PSE)

The three-dimensional reconstructed human psoriatic skin equivalents (PSE) model (SOR-300-FT) obtained using normal epidermal keratinocytes and psoriatic fibroblasts harvested from psoriasis lesions were supplied in inserts of 24-well tissue culture plates (MatTek Corporation, Ashland, MA, USA). Tissue cultures were maintained in SOR-300-FT-MM media and replenished with fresh medium supplemented or not with NPD-0614-13 and NPD-0614-24 every alternate day for 4 days. PSE were processed for H&E staining, morphometry, and immunofluorescence analysis. Supernatants were collected for cytokine evaluation by ELISA assay.

### 4.7. RNA Extraction and Quantitative Real-Time RT-PCR

Total RNA was isolated using the Aurum^TM^ Total RNA Mini kit. Total RNA samples were stored at −80 °C until use. Following DNAse I treatment, cDNA was synthesized using a mix of oligo-dT and random primers and RevertAid^TM^ First Strand cDNA synthesis kit according to the manufacturer’s instructions. Quantitative real time RT-PCR was performed in a total volume of 10 μL with SYBR Green PCR Master Mix and 200 nM concentration of each primer. Sequences of all primers used are shown in Appendix A. Reactions were carried out in triplicates using a CFx96^TM^ Real Time System (Bio-Rad). Melt curve analysis was performed to confirm the specificity of the amplified products. Expression of mRNA (relative) was normalized to the expression of GAPDH mRNA by the change in the ∆ cycle threshold (∆Ct) method and calculated based on 2^−∆Ct^.

### 4.8. RNA Interference Experiments

For the RNA interference experiments, NHKs were transfected with 100 pmol siRNA specific for AhR (si-AhR) [68]. An equivalent amount of non-specific siRNA was used as a negative control. Cells were transfected using Amaxa^®^ human keratinocyte Nucleofector kit, according to manufacturer’s instructions. To ensure identical siRNA efficiency among the plates, cells were transfected together in a single cuvette and plated immediately after nucleofection. Twenty-four hours following transfection, treatments were added to some samples in agreement with the experimental design.

### 4.9. Western Blot Analysis

Cells were lysed in denaturing conditions supplemented with a protease inhibitor cocktail (Roche, Mannheim, Germany), then sonicated. Total cell lysates were clarified by centrifugation at 12,000 rpm for 10 min at 4 °C and then stored at −80 °C until analysis. Following spectrophotometric protein measurement, equal amounts of protein were resolved on acrylamide SDS-PAGE, transferred onto nitrocellulose membrane (Amersham Biosciences, Milan, Italy). Protein transfer efficiency was checked with Ponceau S staining (Sigma-Aldrich). After washing with PBS, the membranes were blocked with 5% fat-free dry milk in PBS with 0.05% Tween-20 for 1 h at room temperature and then treated overnight at 4 °C with primary antibody (according to data sheet instructions). A secondary anti-mouse IgG HRP-conjugated antibody (1:3000) or anti-rabbit IgG HRP-conjugated antibody (1:8000) were used. Antibody complexes were visualized using ECL. A subsequent hybridization with anti-β tubulin (1:10,000) or anti-GAPDH (1:5000) was used as a loading control. The cytoplasm and nuclear fractions were used to determine the level of NF-κB using NE-PER nuclear and cytoplasmic extraction reagents and anti-NF-κB p65 antibody and anti-NF-κB p65 antibody. Protein levels were quantified by measuring the optical densities of specific bands using UVI-TEC System (Eppendorf; Hamburg, Germany). Control value was taken as one fold in each case.

### 4.10. Immunofluorescence Analysis

Cells grown on coverslips previously coated with 2% gelatin on 24-well plates. After treatment cells were fixed in PFA 4% for 15 min at room temperature and immunolabeled with anti-Aryl hydrocarbon Receptor (RPT9) mouse antibody (1:50 in PBS), anti-NF-κB p65 rabbit antibody (1:100 in PBS), and Ki67 (1:300 in PBS). The primary antibodies were visualized using anti-mouse IgG-Alexa Fluor 488 conjugated antibody (1:800 in PBS) and anti-rabbit IgG-Alexa Fluor 555 conjugated antibody (1:800 in PBS) (Cell Signaling). Nuclei were visualized with DAPI (Sigma-Aldrich Srl). Fluorescence signals were analyzed by recording stained images using a cooled CCD color digital camera (Zeiss, Oberkochen, Germany). Quantitative analysis of fluorescence signals was performed using the Zen 2.6 (blue edition) software (Zeiss), evaluating at least 150 cells for each condition, randomly taken from 20 different microscopic fields in three distinct experiments. For AhR fluorescence signals, values are expressed as fold increase of nuclear fluorescence intensity with respect to those obtained in untreated cells and are reported as mean value ± standard deviation (SD) (%). For phospho-NFκB p65 fluorescence immunostaining, results are expressed as the number of cells with a nuclear positive staining and are reported as mean value ± SD (%). For cell proliferation analysis the number of Ki67 positive cells was evaluated counting at least 300 cells randomly taken from 10 different microscopic fields.

### 4.11. Fluorescence Measurement of ROS Levels

Cells were treated with TNF-α (20 ng/mL) or LPS (10 µg/mL) in presence or not with NPD-0614-13 and NPD-0614-24 for 30 min. Intracellular ROS were determined with DCFH_2_-DA. After 20 min incubation with 10 µM DCFH_2_-DA, the signal of DCF (the oxidation product of DCFH_2_-DA) was quantified using the fluorescence multi-plate reader DTX 880 Multimode Detector (Beckman Coulter Srl, Milan, Italy) with an excitation wavelength of 485 nm and emission wavelength of 530 nm.

### 4.12. Protein Determination by Sandwich Enzyme-Linked Immunosorbent Assay (ELISA)

Culture supernatants were collected and centrifuged to remove cell detritus. Aliquots were stored at −80 °C until use. IL-6 and IL-8 protein levels were determined using commercially available ELISA kit, according to the manufacturer’s instructions. The results were normalized for the number of cells contained in each sample. The measurement was performed in duplicate for each sample. The absorbance at 450 nm was recorded using a DTX880 Multimode Detector spectrophotometer (Beckman Coulter, Milan, Italy).

### 4.13. Histology, Morphometry, and Immunofluorescence Analysis of HEEs and PS-HEEs

For histological and morphometric analyses, de-paraffinized sections HEEs were stained with hematoxylin and eosin (H&E). Serial sections were analyzed by recording stained images using a cooled CCD color digital camera (Zeiss, Oberkochen, Germany). Zen 2.6 software (blue edition) (Zeiss) was used for the evaluation of epidermal and stratum corneum thickness. At least 50 measurements were made, for the epidermis and for the stratum corneum, on the images taken under the different experimental conditions. The results are expressed as an average thickness value ± SD. For immunofluorescence analysis sections were dewaxed in xylene, and rehydrated through graded ethanol to PBS. The antigen retrieval was achieved by heating sections at pH6. Then sections were blocked for 15 min with 5% normal goat serum in PBS and incubated overnight at 4 °C with the following primary antibodies: anti-involucrin (1:200 in PBS) (AbCam), anti-filaggrin (1:500 in PBS) (15C10; Monosan, Sanbio, Uden, Netherlands), anti-psoriasin (S100A7) (1:300 in PBS) (Sigma), and anti-beta defensin 2 (1:1000 in PBS) (AbCam). The primary antibodies were visualized incubating sections for 2 h at room temperature with the following secondary antibodies: anti-mouse IgG-Alexa Fluor 488 conjugated antibody (1:800 in PBS) and anti-rabbit IgG-Alexa Fluor 594 conjugated antibody (1:800 in PBS) (Cell Signaling). Sections were mounted using ProLong mounting with DAPI (TermoFisher). Images of fluorescent-stained sections were recorded using a CCD camera on a fluorescent microscope Zeiss (Axioskop 2 Plus) and the intensity of fluorescence signals was quantified using Zeiss Zen 2.6 (blue edition) Software for image analysis. For each experimental sample, sections obtained from three different cutting levels (upper, central, lower) were analyzed in all their length.

### 4.14. Statistical Analysis

Data were represented as mean ± SD of three independent experiments using at least three different donors for primary cells. Statistical significance was assessed using paired Student’s *t*-test. The minimal level of significance was *p* < 0.05.

## Figures and Tables

**Figure 1 ijms-22-07501-f001:**
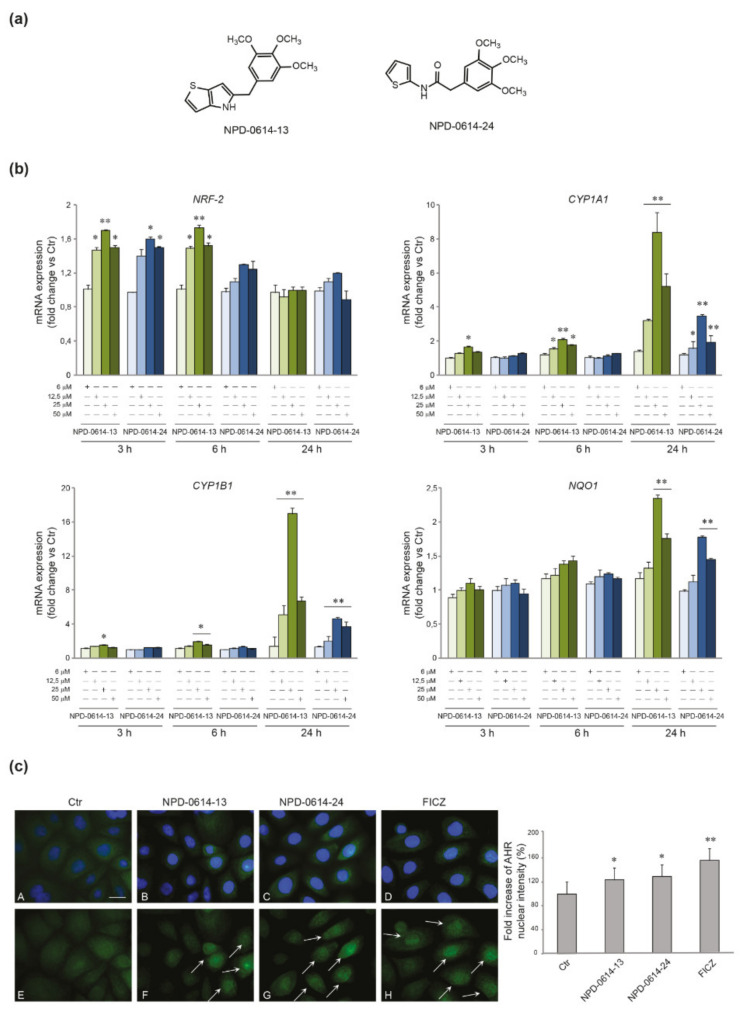
Activation of AhR signaling in response to NPD-0614-13 and NPD-0614-24 in NHKs. (**a**) Schematic representation of the chemical structure of NPD-0614-13 and NPD-0614-24. (**b**) Quantitative real time PCR analysis of NRF-2, CYP1A1, CYP1B1, and NQO1 in NHKs treated with NPD-0614-13 and NPD-0614-24 (6, 12.5, 25, and 50 μM) for 3–6–24 h. All mRNA values were normalized against the expression of GAPDH and were expressed relative to untreated control cells. The data in the graphs are mean ± SD of three independent experiments (* *p* < 0.05, ** *p* < 0.01 vs. untreated control). (**c**) Representative immunofluorescence images and fluorescence quantitative analysis of AhR localization in NHKs untreated (**A**,**E**) and stimulated with NPD-0614-13 (25 μM) (**B**,**F**), NPD-0614-24 (25 μM) (**C**,**G**) and FICZ (100 nM) (**D**,**H**) for 1 h. Following the treatments, AhR signal (green) appeared mainly localized in the nucleus (F,G,H). Arrows indicate the cells with a high nuclear localization of AhR. Nuclei were counterstained with DAPI (A-D). Bar: 20 μm. Results are expressed as fold increase of nuclear fluorescence intensity with respect to the values obtained in untreated cells and are reported as mean value (%) ± SD of three independent experiments (* *p* < 0.05, ** *p* < 0.01 vs. untreated cells).

**Figure 2 ijms-22-07501-f002:**
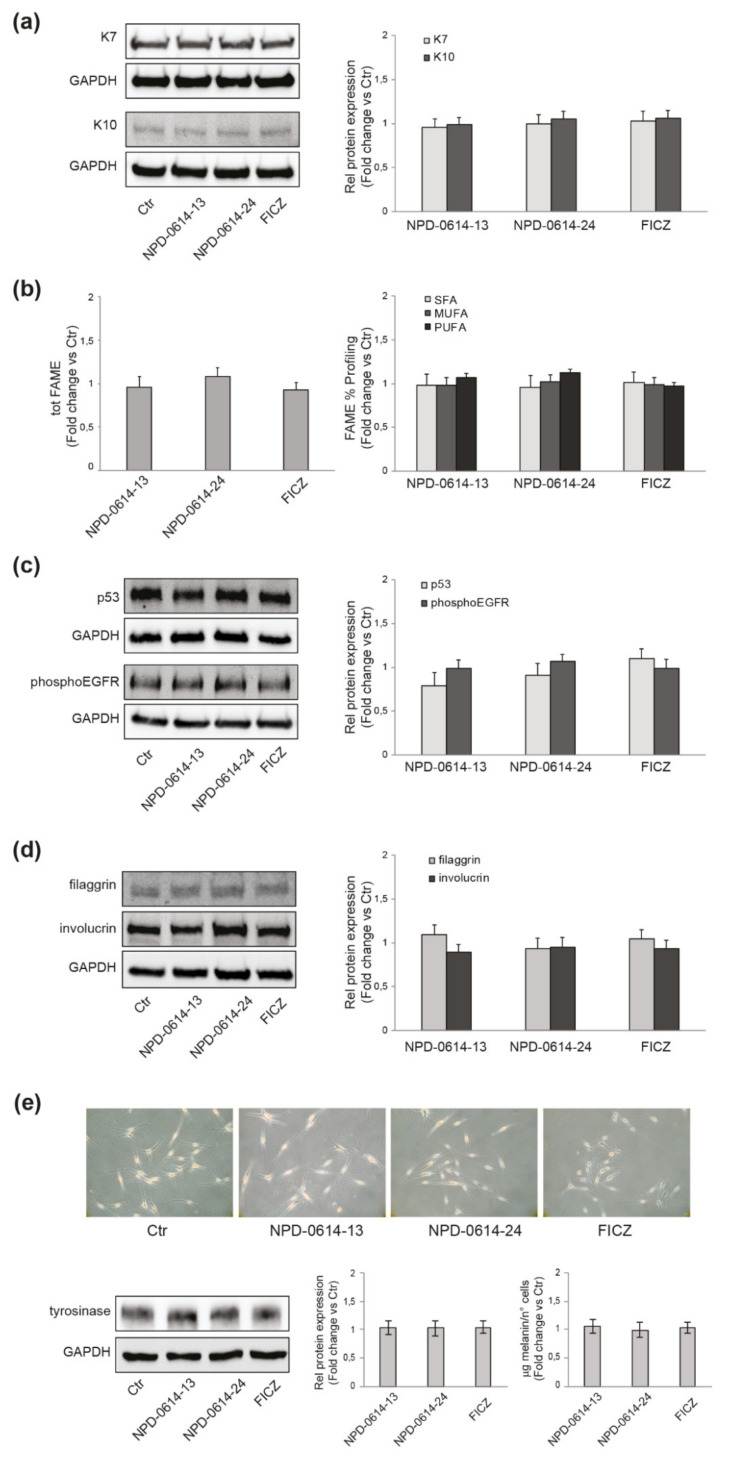
NPD-0614-13 and NPD-0614-24 safety profile in different skin cell populations. (**a**) Western blot and corresponding densitometric analysis of K7 and K10 protein expression in SZ95 sebocytes treated with NPD-0614-13 (25 μM), NPD-0614-24 (25 μM) and FICZ (100 nM) for 48 h. (**b**) Percentage composition of saturated, mono- and polyunsaturated fatty acids in SZ95 sebocytes treated as above for 72 h. (**c**) Western blot and corresponding densitometric analysis of p53 and phospho-EGFR protein expression in NHKs treated as above for 48 h and 5 min, respectively. (**d**) Western blot and corresponding densitometric analysis of filaggrin and involucrin protein expression in NHKs treated as above for 48 h. (**e**) Phase-contrast analysis, western blot with corresponding densitometric analysis of tyrosinase protein expression and melanin content analysis in NHMs treated with NPD-0614-13 (25 μM), NPD-0614-24 (25 μM) and FICZ (100 nM) for 72 h and 5 days, respectively. GAPDH was used as an endogenous loading control for western blot analysis. Results are expressed as the fold change respect to untreated control cells. Data represent the mean ± SD of three independent experiments.

**Figure 3 ijms-22-07501-f003:**
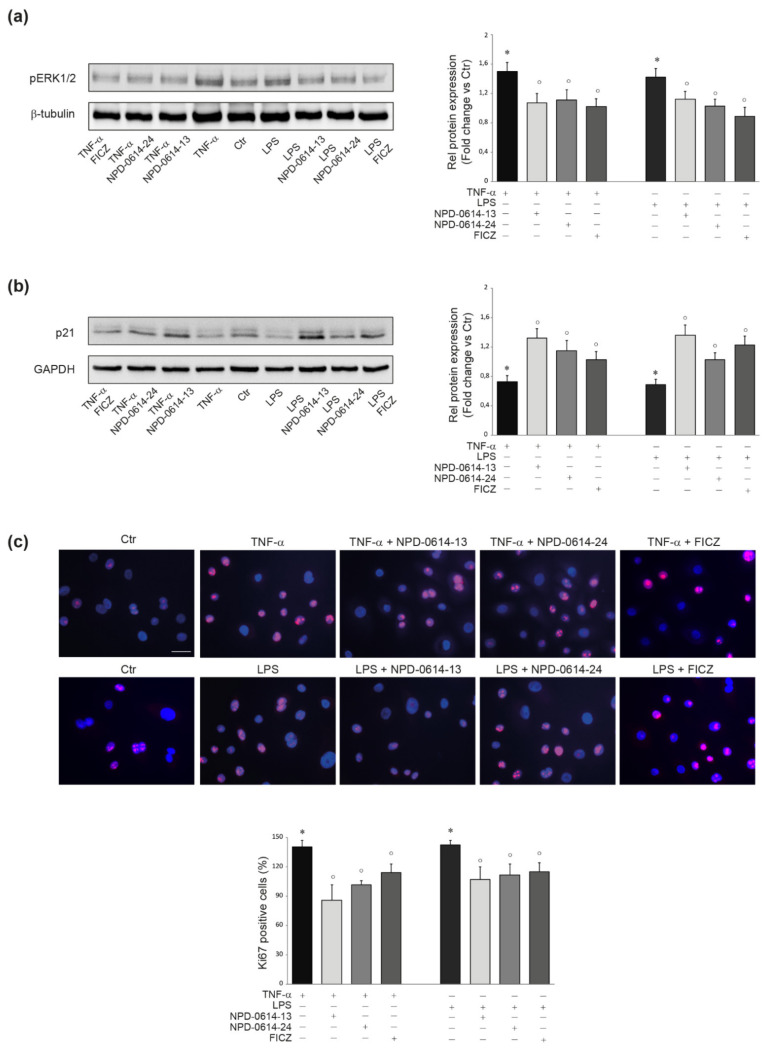
NPD-0614-13 and NPD-0614-24 recovered the altered proliferation induced by TNF-α or LPS in NHKs. (**a**) Western blot analysis and corresponding densitometric analysis of pERK1/2 protein expression in NHKs treated with TNF-α (20 ng/mL) or LPS (10 μg/mL) in the presence or absence of NPD-0614-13, NPD-0614-24 (25 μM) and FICZ (100 nM) for 1 h. (**b**) Western blot analysis and corresponding densitometric analysis of p21 protein expression in NHKs treated as above for 6 h. β-tubulin or GAPDH were used as endogenous loading control. Results are expressed as the fold change respect to untreated control cells. Data represent the mean ± SD of three independent experiments (significance vs. untreated control or vs. stimulated cells are marked with * and °, respectively; * *p* < 0.05 vs. untreated cells; ° *p* < 0.05 vs. TNF-α or LPS stimulated cells). (**c**) Representative immunofluorescence images and fluorescence quantitative analysis of Ki67 expression in NHKs treated as above for 48 h. Nuclei were counterstained with DAPI. Bar: 20 μm. Results are expressed as the mean percentage of Ki67-positive cells (* *p* < 0.05 vs. untreated cells; ° *p* < 0.05 vs. TNF-α or LPS stimulated cells).

**Figure 4 ijms-22-07501-f004:**
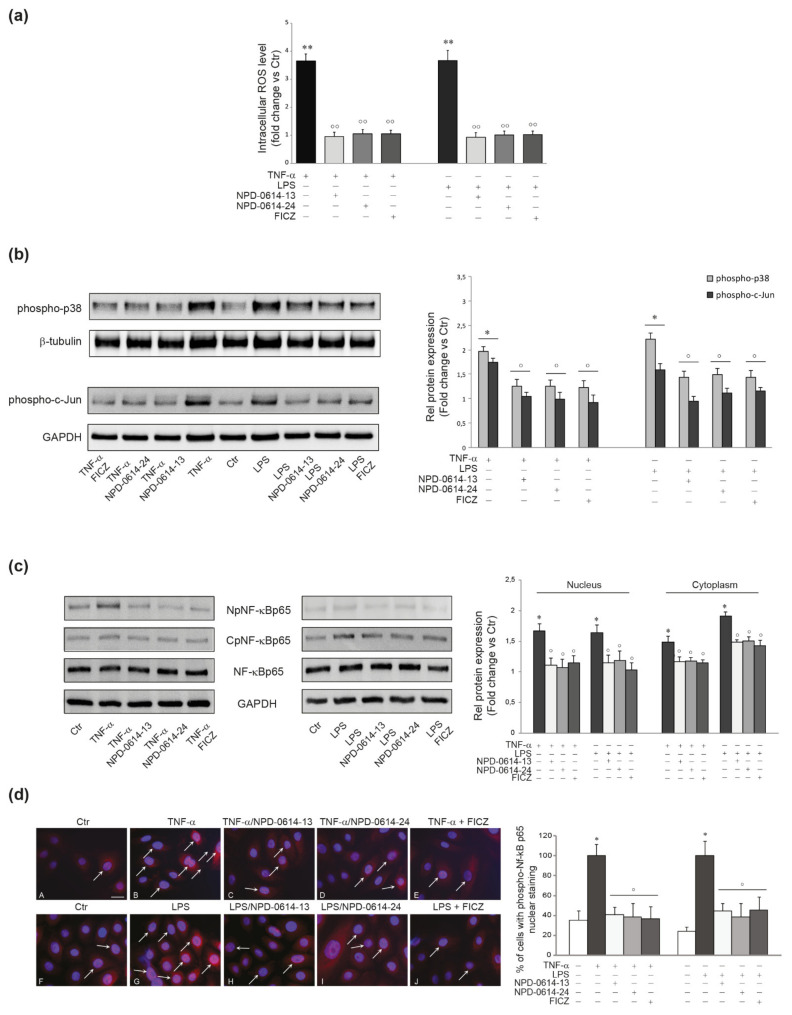
NPD-0614-13 and NPD-0614-24 counteracted the inflammatory pathways induced by TNF-α or LPS in NHKs. (**a**) ROS levels in NHKs stimulated with TNF-α (20 ng/mL) or LPS (10 μg/mL) in the presence or absence of NPD-0614-13, NPD-0614-24 (25 μM) or FICZ (100 nM) for 30 min. Results are expressed as the fold change respect to untreated control cells (significance vs. untreated control or vs. stimulated cells are marked with * and °, respectively; ** *p* < 0.01 vs. untreated cells; °° *p* < 0.01 vs. TNF-α or LPS stimulated cells). (**b**) Western blot analysis and corresponding densitometric analysis of phospho-p38 and phospho-c-Jun protein expression in NHKs treated as above for 30 min. β-tubulin and GAPDH were used as endogenous loading control. Results are expressed as the fold change respect to untreated control cells (* *p* < 0.05 vs. untreated cells; ° *p* < 0.05 vs. TNF-α or LPS stimulated cells). (**c**) Western blot analysis and corresponding densitometric analysis of NF-κB p65 at nucleus (NpNF-κBp65) and cytoplasm level (CpNF-κBp65) on NHKs stimulated as above for 30 min. GAPDH was used as the loading control. Results are expressed as the fold change respect to untreated control cells (* *p* < 0.05 vs. untreated cells; ° *p* < 0.05 vs. TNF-α or LPS stimulated cells). (**d**) Representative immunofluorescence images and fluorescence quantitative analysis of NF-κB p65 localization in NHKs untreated (**A**,**F**) and stimulated with TNF-α (20 ng/mL) (**B**) and LPS (10 μg/mL) (**G**), in the presence or absence of NPD-0614-13 (25 μM) (**C**,**H**), NPD-0614-24 (25 μM) (**D**,**I**) and FICZ (100 nM) (**E**,**J**) for 30 min. Nuclei were counterstained with DAPI. Bar: 20 μm. Results are expressed as the number of cells with a nuclear positive staining and are reported as mean value ± SD (%) (* *p* < 0.05 vs. untreated cells; *p* < 0.05 vs. TNF-α or LPS stimulated cells).

**Figure 5 ijms-22-07501-f005:**
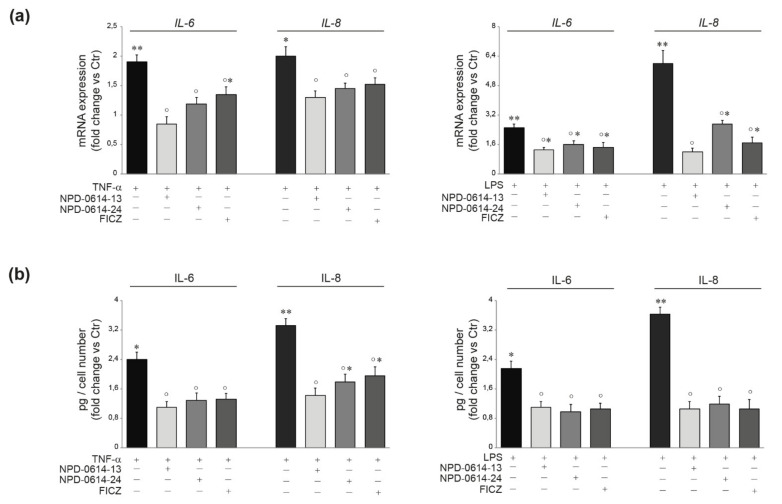
NPD-0614-13 and NPD-0614-24 counteracted the TNF-α or LPS induced expression of pro-inflammatory cytokines in NHKs. (**a**) Quantitative real time PCR analysis of IL-6 and IL-8 in NHKs treated with TNF-α (20 ng/mL) or LPS (10 μg/mL) in the presence or absence of NPD-0614-13, NPD-0614-24 (25 μM) and FICZ (100 nM) for 6 h. All mRNA values were normalized against the expression of GAPDH and were expressed relative to untreated control cells. (**b**) IL-6 and IL-8 quantitation by ELISA in NHKs treated as above for 24 h. The data are expressed as mean ± SD of three independent experiments (significance vs. untreated control or vs. stimulated cells are marked with * and °, respectively; * *p* < 0.05, ** *p* < 0.01 vs. untreated control; ° *p* < 0.05 vs. TNF-α or LPS stimulated cells).

**Figure 6 ijms-22-07501-f006:**
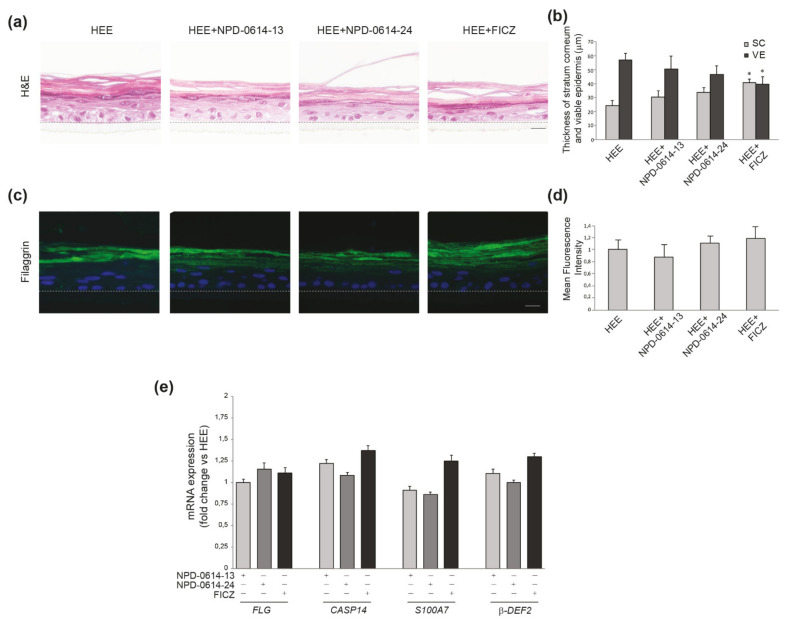
NPD-0614-13 and NPD-0614-24 safety profile on human epidermal equivalents (HEEs). (**a**) Hematoxilin–eosin staining of HEEs cultured in presence or absence of NPD-0614-13, NPD-0614-24 (25 μM) and FICZ (100 nM) for 72 h. Bar: 50 μm. (**b**) Morphometric analysis of the stratum corneum (SC) and the viable epidermal layers (VE). The thickness of SC and VE was expressed as mean value ± SD (* *p* < 0.05 vs. untreated HEEs). (**c**) Representative immunofluorescence images after using filaggrin antibody. Nuclei were counterstained with DAPI. Bar: 20 μm. (**d**) Quantitative analysis of fluorescence intensity was expressed as mean value ± SD. Dashed black/white lines represent the junction between the basal layer and the membrane of the insert. (**e**) Quantitative real time PCR analysis of filaggrin, caspase-14, S100A7 and beta-defensin-2 in HEEs, in the presence of absence of NPD-0614-13, NPD-0614-24 (25 μM) and FICZ (100 nM) for 72 h. All mRNA values were normalized against the expression of GAPDH and were expressed relative to untreated HEEs.

**Figure 7 ijms-22-07501-f007:**
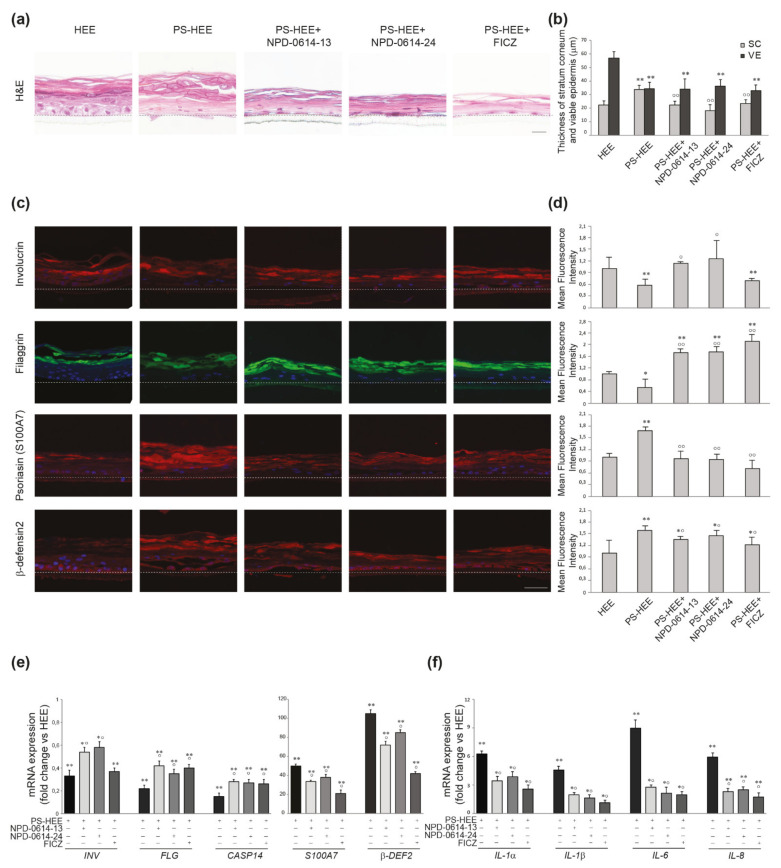
Effects of NPD-0614-13 and NPD-0614-24 on psoriasis-like human epidermal equivalents (PS-HEEs). (**a**) Hematoxilin–eosin staining of HEEs and PS-HEEs cultured in the presence or absence of NPD-0614-13, NPD-0614-24 (25 μM) and FICZ (100 nM) for 72 h. Bar: 50 μm. (**b**) Morphometric analysis of stratum corneum (SC) and viable epidermal layers (VE). The thickness of SC and VE was expressed as mean value ± SD (significance vs. untreated HEEs or vs. PS-HEEs are marked with * and °, respectively; ** *p* < 0.01 vs. untreated HEEs; *p* < 0.01 vs. PS-HEEs). (**c**) Representative immunofluorescence images after using antibodies against involucrin, filaggrin, S100A7, and beta-defensin-2. Nuclei were counterstained with DAPI. Bar: 20 μm. (**d**) Quantitative analysis of fluorescence intensity was expressed as mean value ± SD (* *p* < 0.05, ** *p* < 0.01 vs. untreated HEEs; ° *p* < 0.05, °° *p* < 0.01 vs. PS-HEEs). Dashed black/white lines represent the junction between the basal layer and the membrane of the insert. Quantitative real time PCR analysis of the expression of (**e**) involucrin, filaggrin, caspase-14, S100A7, beta-defensin 2, and (**f**) IL-1α, IL-1β, IL-6, and IL-8 in PS-HEEs in the presence of absence of NPD-0614-13, NPD-0614-24 (25 μM), or FICZ (100 nM) for 72 h. All mRNA values were normalized against the expression of GAPDH and were expressed relative to untreated HEEs (* *p* < 0.05, ** *p* < 0.01 vs. HEEs; *p* < 0.05 vs. PS-HEEs).

**Figure 8 ijms-22-07501-f008:**
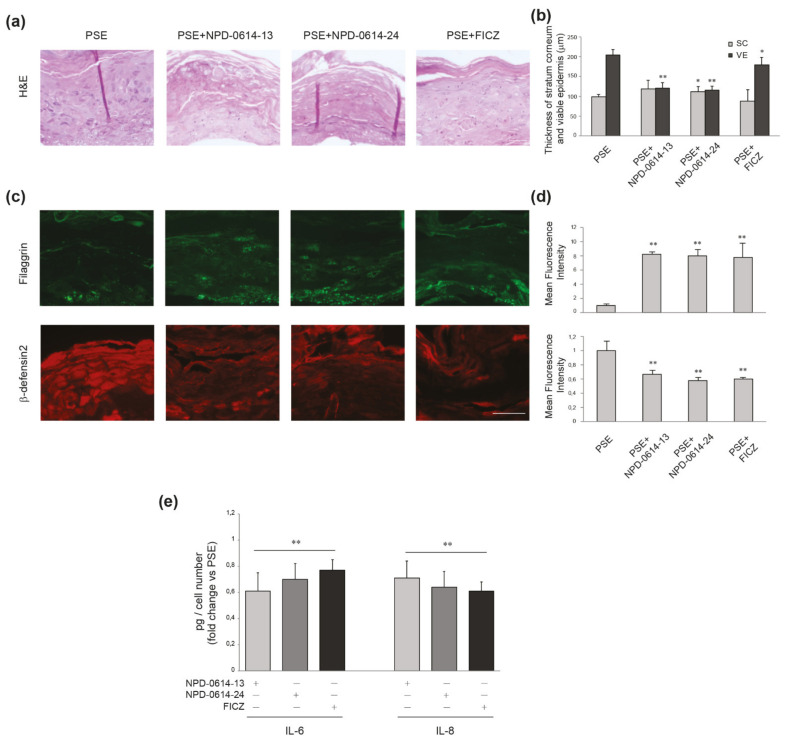
Effects of NPD-0614-13 and NPD-0614-24 on psoriasis skin equivalents (PSEs). (**a**) Hematoxilin–eosin staining of PSEs cultured in the presence or absence of NPD-0614-13, NPD-0614-24 (25 μM) and FICZ (100 nM) for 4 days. Bar: 50 μm. (**b**) Morphometric analysis of stratum corneum (SC) and viable epidermal layers (VE). The thickness of SC and VE was expressed as mean value ± SD (* *p* < 0.05, ** *p* < 0.01 vs. PSEs). (**c**) Representative immunofluorescence images after using antibodies against filaggrin and beta-defensin-2. Nuclei were counterstained with DAPI. Bar: 50 μm. (**d**) Quantitative analysis of fluorescence intensity was expressed as mean value ± SD (** *p* < 0.01 vs. PSEs). (**e**) IL-6 and IL-8 quantitation by ELISA in PSEs in the presence of absence of NPD-0614-13, NPD-0614-24 (25 μM) or FICZ (100 nM) for 72 h. The data are expressed as mean ± SD (** *p* < 0.01 vs. PSEs).

## Data Availability

The data that support the findings of this study are available on request from the corresponding author.

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
