# Peer review of "Anti-Inflammatory and Pro-Differentiating Properties of the Aryl Hydrocarbon Receptor Ligands NPD-0614-13 and NPD-0614-24: Potential Therapeutic Benefits in Psoriasis"

_ijms, 2021, doi:10.3390/ijms22147501_

Round 1

Reviewer 1 Report

This is a study on the anti-inflammatory effect of 2 ligands of aryl hydrocarbon receptors in keratinocytes, melanocytes, and reconstituted skin models. The novelty of this study is not very high as studies on the effects of these two drugs on relieving gut inflammation have already been reported. However, attempts to apply it to skin psoriasis are worthy of evaluation.

  1. Please describe how the drug was discovered and selected in the Introduction.
  2. Present numerical data on the receptor binding affinity of these two ligands. Comparison with existing drugs is necessary.
  3. Figure data are not read well. Please increase the font and resolution.
  4. Please write the full name first before the abbreviation. Please write the full name of the receptor in the title.
  5. Please compare the pros and cons of these two ligands in the Discussion.
  6. The biggest drawback of this study is the lack of in vivo data. Supplementation or discussion is necessary for this point.

Reviewer 2 Report

A nice manuscript, studying the role of two novel AHR agonists for anti-inflammatory therapy of skin diseases. 

If the compounds induce AHR and this leads to CYP1A1 and CYP1B1 induction (Fig. 1), are these effects induced by FICZ?

In an in vivo situation, AHR ligands would affect other cells in the skin. The authors should show, how the most common immune cell types involved in psoriasis respond to the AHR induction.

Minor comment - the quality (resolution) of the figures was very low, this should be addressed in the final submission.

Round 2

Reviewer 2 Report

Thank you for the response, good luck with the manuscript.